# Preparation and Characterization of Water-borne Polyurethane Based on Benzotriazole as Pendant Group with Different N-Alkylated Chain Extenders and Its Application in Anticorrosion

**DOI:** 10.3390/molecules27217581

**Published:** 2022-11-04

**Authors:** Aamna Bibi, Ethan Tsai, Yun-Xiang Lan, Kung-Chin Chang, Jui-Ming Yeh

**Affiliations:** Department of Chemistry, Center for Nanotechnology and R & D Center for Membrane Technology, Chung-Yuan Christian University, Chung Li 32023, Taiwan

**Keywords:** water-borne polyurethane, benzotriazole, N-alkylated chain extenders, anti-corrosion

## Abstract

A series of novel anti-corrosive coatings were synthesized successfully. Water-borne polyurethane (WPU) was synthesized using polyethylene glycol and modified by grafting benzotriazole (BTA) as a pendant group (WPU-g-BTA) and N-alkylated amines (ethylene diamine (A), diethylene triamine (B), triethylene tetramine (C)) as side-chain extenders. Fourier-transform infrared spectroscopy, thermogravimetry, and dynamic mechanical analyses were used to characterize the structural and thermomechanical properties of the samples. A gas permeability analyzer (GPA) was used to evaluate molecular barrier properties. The corrosion inhibition performance of WPU-g-BTA-A, WPU-g-BTA-B, and WPU-g-BTA-C coatings in 3.5 wt% NaCl solution was determined by electrochemical measurements. WPU-g-BTA-C coating synthesized with a high cross-linking density showed superior anticorrosive performance. The as-prepared coatings exhibited a very low icorr value of 0.02 µA.cm^−2^, a high Ecorr value of −0.02 V, as well as excellent inhibition efficiency (99.972%) and impedance (6.33 Ω) after 30 min of exposure.

## 1. Introduction

Corrosion is responsible for economic losses, and time consumption is one of the major concerns in architectural engineering. Polymer paints and coatings, such as epoxy, polyurethane, polyacrylate, alkyd, polyester, and phenolic resins are the simplest and cheapest compounds to protect against corrosion [1,2]. Polyurethanes (PUs) are one of the popular materials used in different applications, such as coatings, adhesives, sealants, elastomers, foams, etc. [3]. Recently, water-borne polyurethanes (WPU) have gained research attention due to their unique properties of environment friendliness and easy processing [4]. WPUs are used in strain sensors [5], implantable biomaterials [6], cancer therapy [7], anti-bacterial agents [8], and coatings, adhesives, paints, leather, and composites.

Yeh et al. developed a WPU/Na+ -montmorillonite (Na+ -MMT) clay nanocomposite coating for practical evaluation of corrosion protection efficiency [9]. In 2019, Hou et al. reported WPU nanocomposites with complex packing of GO and CB as anti-corrosive coating [10]. Liu et al. proposed WPU nanocomposites with polyether-amine functionalized graphene oxide for anti-corrosion [11]. Bilal et al. and Xie et al. investigated the anti-corrosive properties of WPU reinforced with 2D nanosheets of graphene, GO, hexagonal boron nitride [12,13], and Ti3C2 MXenes [14]. Many other researchers developed WPU nanocomposites for anti-corrosive coatings by using graphene/graphene oxide [13], functionalized nanosilica [15], mGO/PTFE [16], TiO_2_/PANI/HNT/CNT [17], and MWCNTs [18] and by incorporating a cyclic rigid ring [19]. However, mechanical damages to these coating can create weak points, leading to various types of corrosion including general, flow assisted, galvanic, and pitting [20,21]. In this regard, protection coatings require the use of corrosive inhibitors to solve these issues and enhance anti-corrosion performance. Corrosion inhibitors generally contain specific characters such as nitrogen, sulphur, oxygen, and so on, or some special unsaturated functional groups, etc. They have the advantages of simple operation, significant corrosion protection effect, and small dosage. Therefore, they have received extensive attention from corrosion protection scientists [22,23,24].

For many years, hexavalent chromates were used as traditional inhibitors, but their application was restricted due to their toxic nature. Recently, heterocyclic molecules, such as benzotriazole (BTA), have been widely used as corrosion inhibitors [25,26].

In 2014, Xuewu et al. reported the excellent corrosion protection of BTA-loaded raspberry-like hollow microspheres with and without WPU films [26], in which copper electrode coated by WPU film containing BTA-loaded raspberry-like hollow microspheres showed a corrosion inhibition efficiency of 84.4% after 30 h of immersion in alkaline meium (pH = 10). Abdullayev et al. studied polyurethane and acrylic paints doped with halloysite loaded with benzotriazole, 2-mercaptobenzimidazole, and 2-mercaptobenzothiazole as corrosion inhibitors [27]. Li et al. reported different smart sensing coatings such as PDVB-graft-P(DVB-co-AA)-BTA, Halloysite-BTA, and IPDI@PU/PUF for the evaluation of corrosion inhibition efficiency [28]. Scholars have also developed systems based on silica by using nano-containers, nano-capsules, and microspheres loaded with BTA for corrosion inhibition [29,30].

Coatings also undergo photo-oxidative degradation upon UV exposure. In this regard, various strategies are used, including the addition of organic UV absorbers (UVA) and hindering amine light stabilizers or functional particles (TiO_2_, ZnO) [31,32,33,34]. Common chemical classes of organic UVA are benzophenones, benzotriazoles, triazines, malonates, and oxalanilides [35].

Khannna et al. [36] and Rabello et al. [37] used nano ZnO and hydroxyphenyl triazine as UV stabilizers in polymer coatings. Campbell et al. used ZnO/TiO_2_ [38], and Adomsons reported the use of benzotriazole and triazine [39] as UVA in automotive plastic coating. The introduction of side chain improves the properties and flexibility of the WPU main chain [40]. In particular, methyl group [41], long-branched fatty acids [42], organo-phosphorous groups [43], and siloxane groups [44] have been used as side chains. The effects of chain extenders on the adhesion and mechanical properties of WPU films have also been studied [45]. Lu et al. reported a series of WPU with 3-(2-aminoethylamino)propyldimethoxymethylsilane (APTS) as chain extender [46]. Lee et al. reported WPU with cystamine as chain extender [47]. In 2016, Fan et al. synthesized and characterized WPU conjugated with diol bearing a cyclic phosphoramidate pendant group, namely, 2-(5,5-dimethyl-2-oxo-2l5-1,3,2-dioxaphosphinan-2-ylamino)-2-methyl-propane-1,3-diol (PNMPD) [48]. Moreover, WPU with diol as an extender with a triethoxysilane group [49] and alkoxysilane side group [50] was developed.

In this study, we aimed to synthesize an anti-corrosive coating by grafting BTA onto the polymer backbone via a covalent linkage. WPU was prepared using BTA-based Eversorb 80 (EV 80) as a pendant group with HDI (trimer) N3300 as a bridging agent. Novel WPU-g-BTA films were then synthesized with different N-alkylated amines as chain extenders. The structure of WPU-g-BTA was tested by FTIR. Thermogravimetric analysis (TGA), dynamic mechanical analysis (DMA), and gas permeability analysis (GPA) were conducted to investigate the thermomechanical and molecular barrier properties of the synthesized coatings. Anti-corrosive properties were analyzed by potentiodynamic polarization (PDP) and electrochemical impedance spectroscopy (EIS) in a corrosive environment (3.5 wt% NaCl).

## 2. Results and Discussion

### 2.1. Characterization

The as-synthesized compound was characterized through FTIR analysis. The representative IR spectra of EV 80, HDI trimer, and WPU-g-BTA are shown in Figure 1. The characteristic band for the NCO- group of HDI trimer at 2260 cm^−1^ disappeared in the corresponding polymer. The absorption band at 3338 cm^−1^ indicated the hydrogen-bonded N–H group of the polymer. The absorption band for the C–H group was detected at 2870 cm^−1^. The strong band at 1638 and 1690 cm^−1^ were assigned to the C=O groups.

### 2.2. Gel Contents

The as-synthesized anti-corrosive coatings had different gel contents (%). Gel content is correlated with crosslinking density [51] (Figure 2).

For example, WPU-g-BTA-A had 13.57% gel, which was increased to 18.79% with ‘B’ as the chain extender. Meanwhile, WPU-g-BTA-C showed the highest gel content (21.68%) with the longest chain length (Table 1).

### 2.3. Electrochemical Corrosion Measurement

#### 2.3.1. Potentiodynamic Polarization (PDP) Measurement

PDP analysis was used for quantitative measurements of the inhibition efficiency (IE) of WPU-g-BTA-A, WPU-g-BTA-B, and WPU-g-BTA-C coatings under 3.5 wt% NaCl environment for 30 min and three days respectively.

##### Short Term Corrosion Study (30 Min)

The thickness of the coating was set at 060 ± 2 µm. The Tafel plots for the uncoated and coated CRS are presented in Figure 3 and the kinetic corrosion parameters including corrosion potential (Ecorr), corrosion current density (icorr), and IE (%) are summarized in Table 2.

The inhibition efficiency (IE) of the as-prepared coatings was calculated using Equation (1) [15]:i_corr_(0) − i_corr_ (i) × 100i_corr_(0)(1)
where i_corr_(0) and _icorr_(i) denote the corrosive current densities for the uncoated and coated CRS electrodes, respectively.

Ecorr and icorr are substantial parameters that measure the tendency for corrosion and corrosion inhibition as the corrosion rate is directly proportional to Ecorr and inversely proportional to icorr. Thus, the coating with the lowest icorr and the highest Ecorr exhibited the best anticorrosive performance. The coated CRS electrode demonstrated better corrosion protection than the uncoated sample (Figure 3).

The Ecorr and icorr values of the uncoated CRS (black curve) were −0.811 V and 71.77 µA.cm^−2^, respectively. The icorr value decreased (0.079 µA.cm^−2^) in WPU-g-BTA-A (red curve), which exhibited excellent barrier property. With respect to CRS, a drastic shift in Ecorr to more positive potential (−0.3311 V) was observed. The Ecorr value for WPU-g-BTA-B was further moved to −0.252 V with a smaller icorr value of 0.068 µA.cm^−2^ (green curve). Eventually, CRS coated with WPU-g-BTA-C (blue curve) exhibited a very low icorr value of 0.020 µA.cm^−2^, which is about 4 and 3.4 orders of magnitude lower than those of WPU-g-BTA-A and WPU-g-BTA-B, respectively. However, the Ecorr for the coating was −0.1983 V, which is more positive than that of WPU-g-BTA-A and WPU-g-BTA-B. Hence, CRS coated with WPU-g-BTA blocked the diffusion of electrolytes. The results indicated that the corrosion inhibition performance of WPU-g-BTA coatings increased with increasing cross-linking density from sample A to C. Overall, CRS-coated samples showed significantly higher corrosion inhibition performance than bare CRS. The IE % values are listed in Table 2. The corrosion inhibition efficiency of the coated CRS was higher and increased as the length of the chain extender increased (A to C). The CRS coated with WPU-g-BTA-C showed extremely higher corrosion inhibition efficiency (99.972%). In the present study, the order of inhibition efficiency was found to be
WPU-g-BTA-A < WPU-g-BTA-B < WPU-g-BTA-C

##### Long Term Anti-Corrosion Study (3 Days)

In Figure 4 and Table 3, it can be seen that, compared with the bare CRS, self-corrosion potentials of the samples, which were covered with organic coating, shifted more significantly to the right and their corrosion currents also had a downward trend.

The corrosion potentials of the samples shifted from −0.634 V for the bare steel to −0.633 V for WPU−g−BTA−A, −0.576 V for WPU−g−BTA−B, and −0.550 V for WPU−g−BTA−C. The corrosion current (icorr) decreased from 4.98 µA.cm^−2^ for CRS to 4.17 µA.cm^−2^ for WPU−g−BTA−A, 4.08 µA.cm^−2^ for WPU−g−BTA−B, and 3.41 µA.cm^−2^ for WPU−g−BTA−C, where WPU−g−BTA−C had the smallest corrosion current. Further, it is also noted that the CRS exhibited Rp value of 2.587 kΩ whereas the Rp value for WPU−g−BTA−C coated CRS is 3.127 kΩ. This increase in Rp value clearly indicates that the coating acts as a physical barrier and appreciably resists the corrosion of the CRS from corrosion medium. Moreover, the highest corrosion protection efficiency (P_EF_) % achieved for WPU−g−BTA−C coating was 31.56% after 3 days’ immersion in 3.5 wt% NaCl _(aq)_.

#### 2.3.2. Morphological Studies

The cross-sectional images of before (top) and after (bottom) the corrosion test in the 3.5 wt% NaCl solution for three days are shown in Figure 5. The cross-sectional image of WPU−g−BTA−A in shows that the corroded layer was uniform and without pits. The image of WPU−g−BTA−B in also shows a rough corroded layer with a few pits, while the WPU−g−BTA−C coating shows a relatively less rough surface. The specimens with higher cross-linking density showed complex corrosion morphologies.

SEM images of freshly polished surface (top) as well as specimens retrieved from 3.5 wt% NaCl solution (bottom) after 3 days of immersion are shown in Figure 6. The freshly polished WPU−g−BTA coatings showed a relatively smooth surface as shown in Figure 6a,b, while the longest chain extender C due to high crosslinking density strengthened the coating in WPU−g−BTA−C film showed a coarse surface, as shown in Figure 6c. The WPU−g−BTA specimens retrieved from corrosive media showed a rough surface with corrosion products. The WPU−g−BTA−C coating with high crosslinking density seemed compact, tight, and wrapped around the corrosion products, which is beneficial to shield them from corrosive medium.

#### 2.3.3. Electrochemical Impedance Spectroscopy (EIS)

EIS is an effective technique used to evaluate the anticorrosion performance of as-prepared coatings. The impedance spectrum was recorded on uncoated CRS and CRS coated with WPU−g−BTA−C, WPU−g−BTA−B, and WPU−g−BTA−C after 30 min of immersion in 3.5 wt% NaCl solution (Figure 4 and Table 2) at 298 K.

It can be found from Figure 7 that for CRS, the Nyquist plot consisted of a flattened capacitive loop. The Nyquist plot consisted of a single semicircle over the immersion period, suggesting a complete capacitive and good barrier property. The radius of the semicircle increased with increasing chain length from WPU−g−BTA−A to WPU−g−BTA−C. The radius of the capacitive arc indicated the corrosion rate. The capacitive arc with larger radius had a slower corrosion rate, indicating better corrosive inhibition. The diameter of the Nyquist curve for WPU-g-BTA coatings increased with various chain extenders from A to C. The WPU−g−BTA−C coating had the largest radius, indicating excellent corrosion resistance.

BODE plots are one of the most commonly used tools in corrosion science for monitoring local corrosion damages. Figure 8 is the BODE diagram of coated and un-coated CRS dip in 3.5 wt% NaCl for 30 min at 298 K. A small-amplitude signal was applied over a range of frequencies for EIS measurement. The impedance modulus (log Z) in the BODE plot is usually related to the coatings barrier performance; that is, a large impedance value indicates better corrosion protection. The /Z/ (Table 2) of uncoated CRS was 2.22 Ω, which was increased to 5.99 Ω and 6.21 Ω for WPU−g−BTA−A and WPU−g−BTA−B, respectively. WPU−g−BTA−C, which had the largest impedance modulus, exhibited the best corrosion protection. The trend in the BODE plot was similar to those observed in Tafel and Nyquist plots.

### 2.4. O_2_ Permeability

WPU-g-BTA coatings with film thicknesses of 60 ± 2 µm were used to evaluate molecular barrier properties. The O_2_ gas permeability of the as-prepared coatings with different cross-linking densities at 298 K is presented in Figure 9. The O_2_ permeation across different coatings decreased with changes in the length of the chain extender. Compared with WPU-g-BTA-A, WPU-g-BTA-B film had 60% reduction in gas permeability (Table 1). The best barrier performance was recorded for the WPU-g-BTA-C coating with longer chain length. The incorporation of ‘C’ into the WPU-g-BTA coating with higher cross-linking density can create a tortuous path for O_2_ gas molecules, thereby increasing the total path of the gas. The effect of amine-based chain extenders (A, B, C) with various lengths may result in greater interaction, leading to higher efficiency in restricting the movement of polymer chains [52,53].

### 2.5. Thermomechanical Properties

The thermal degradation behavior is shown in Figure 10. The TGA curves showed the weight loss as a function of temperature under N_2_ atmosphere. All the samples had the same structural units and almost the same degradation pattern. The degradation temperature was slightly higher for WPU−g−BTA−C. The thermograms showed two-step degradation; first, there was a decomposition stage (around 230–270 °C) due to the urethane bond degradation of hard segment; second, there was a degradation of the soft segment polyol. For WPU−g−BTA−C, 5 wt% loss occurred in the range of 222–261.19 °C followed by 215–258 °C for WPU−g−BTA−A and WPU−g−BTA−B (Table 1).

The mechanical properties of the WPU−g−BTA coatings with different cross-linking densities were determined (Table 1). DMA demonstrated the effect of temperature on various anticorrosive coatings. The storage modulus of all the samples is shown in Figure 11.

The modulus of WPU-g-BTA-C was 171.4 MPa, which is higher than those of WPU-g-BTA-A and WPU-g-BTA-B at room temperature (20 °C). The hardening mechanism may be attributed to the various cross-linking density in the given samples. The as-prepared coatings showed a sharply decreased modulus. However, the WPU-g-BTA-C coating had a slightly reduced decreasing tendency of storage modulus, providing a good thermomechanical stability up to 50 °C. As shown in Table 1, the 100% modulus, stress at break, and elongation at break of the WPU-g-BTA films increased with increasing chain length of the extenders. The tensile–strain behavior showed a stress of 51.2–47.56 kgf/cm^2^, elongation at break of 178.85%–283.41%, and tensile modulus of 42.2–46.5 kgf/cm^2^. This increase in properties may be associated with the increased cross-linking density due to the WPU-g-BTA chain end reaction with A, B, and C.

## 3. Experimental

### 3.1. Materials

HDI trimer (N3300) was purchased from Covestro (DESMODUR^®^) (Taoyuan, Taiwan). Eversorb 80 (EV 80) was obtained from Everlight Chemical (Taipei, Taiwan). Polyethylene glycol (PEG, Mn = 300, Sigma Aldrich, St. Louis, MO, USA), acetone, DMBA 2,2-Bis(hydroxymethyl)butyric acid (Uni-ONWARD corporation (TAIPEI, Taiwan)), N, N-dimethyl isopropyl amine (DMIPA Fluka), diethyl amine (A, Fluka), diethyelenetriamine (B, Fluka), and triethyelentetramine (C, Fluka) were used as received. Fourier transform infrared (FTIR) analysis was performed at room temperature on JASCO FT/IR-4100 (Oklahoma City, OK, USA). TGA was performed on DuPont TA Q50 (Boston, MA, USA) in air, and DMA was conducted on Dupont TA Q800. Potentiodynamic Polarization (Tafel). Electrochemical impedance and BODE plots were determined on Autolab (PGSTAT302 N) potentiostatic/galvanostatic analyzer. A GTR 10 gas permeability analyzer (Yanagimoto Co., Kyoto, Japan) was used in the permeation of oxygen gas.

### 3.2. Gel Content

The amount of crosslinking can be measured by a standard tetrahydrofuran insoluble test. The insoluble tetrahydrofuran (THF) of the polyurethane dispersion was measured by mixing 1 g of the polyurethane dispersion with 30 g of THF in a pre-weighed centrifuge tube. After the solution was centrifuged for 2 h at 17,000× *g* rpm, the top liquid layer was poured out and the non-dissolved gel in the bottom was left. The centrifuge tube with the non-dissolved gel was re-weighed after drying in an oven at 110 °C for 2 h.
$$\%\mathrm{THF}\;\mathrm{insoluble}\;\mathrm{of}\;\mathrm{WPU}=\frac{\mathrm{Weight}\;\mathrm{of}\;\mathrm{tube}\;\mathrm{and}\;\mathrm{non}-\mathrm{dissolved}\;\mathrm{gel}-\mathrm{wight}\;\mathrm{of}\;\mathrm{the}\;\mathrm{tube}}{\mathrm{sample}\;\mathrm{weight}*\mathrm{WPU}\;\mathrm{solid}\%}$$

### 3.3. Synthesis of WPU-g-BTA

A 500 mL round-bottom flask with a magnetic stirrer, condenser, and nitrogen inlet was used as reactor. EV 80 (0.1 mole) and HDI trimer (0.3 mole) were first charged and reacted at about 75 °C for 3 h to obtain NCO-terminated prepolymer-1. PEG-300 (0.07 mole) and DMBA (0.03 mole) were placed in the flask and heated for 3 h (prepolymer-2) at 70 °C. Acetone was then added at 40 °C to reduce the viscosity of the prepolymer-2. Then the prepolymers were cooled to 60 °C and neutralized with DMIPA for 30 min. An aqueous dispersion was obtained by adding water (35 °C) to the mixture. Then, the modified HEC was added and stirred for about one hour to homogenize the mixture. Subsequently, amine chain extenders (A, B, C) were added and stirred for the next 1.5 h. Then the mixture was cast onto a Teflon plate and partially dried for two days at 35 °C. Finally, the film was dried for two days at 70 °C. The reaction procedure is shown in Figure 12 [51].

## 4. Conclusions

WPU was prepared by covalent grafting of BTA on the polymer backbone as a pendant group (WPU-g-BTA). Using this WPU-g-BTA, we developed a series of WPU-g-BTAs containing N-alkylated amine cross-links (WPU-g-BTA-A, WPU-g-BTA-B, and WPU-g-BTA-C). The successful synthesis of WPU-g-BTAs was confirmed by FTIR. The following conclusions were drawn:The gel content of WPU-g-BTAs increased from 13.57% to 21.68% with increasing chain length from WPU-g-BTA-A to WPU-g-BTA-C. Further, increasing the chain length of WPU-g-BTAs significantly increased the crosslink density, thereby improving the thermal and mechanical properties.WPU-g-BTAs improved the corrosion resistance performance of the as synthesized material. The superior inhibition efficiency of 99.972% and 31.16% was achieved by WPU-g-BTA-C after 30 min and 3 days of immersion in 3.5 wt% NaCl _(aq)_ respectively. Hence, WPU-g-BTA coatings are considered promising materials for anti-corrosion applications.

## Figures and Tables

**Figure 1 molecules-27-07581-f001:**
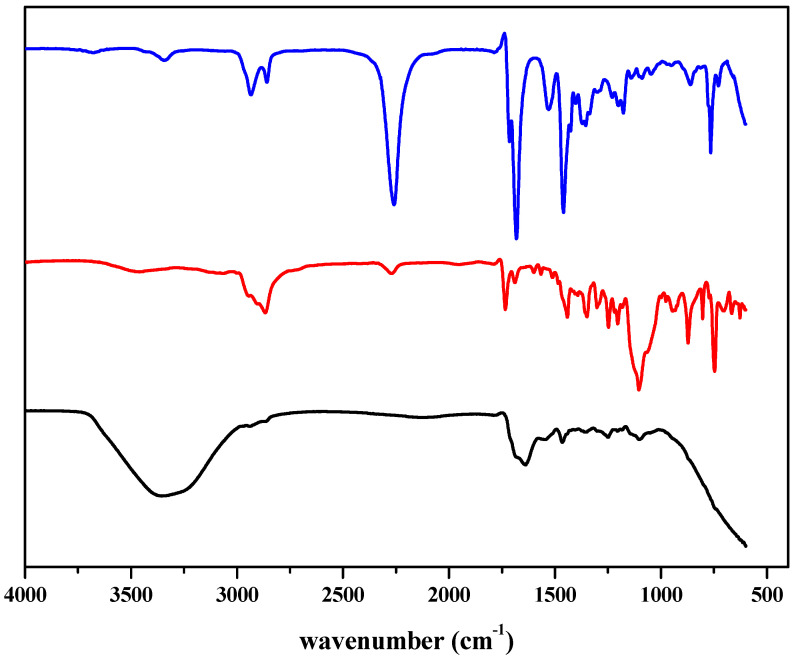
Representative FTIR spectra of (blue) HDI Trimer (red) EV 80 (black) WPU-g-BTA.

**Figure 2 molecules-27-07581-f002:**
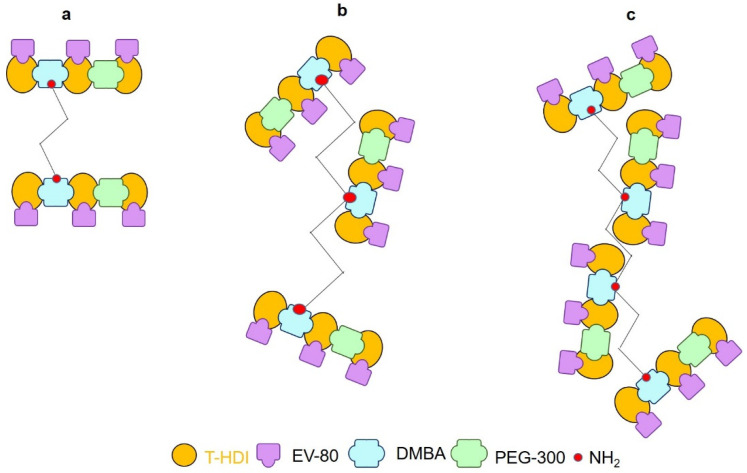
Cross linking density of as prepared (**a**) WPU-g-BTA-A (**b**) WPU-g-BTA-B, and (**c**) WPU-g-BTA-C with different amine chain extenders.

**Figure 3 molecules-27-07581-f003:**
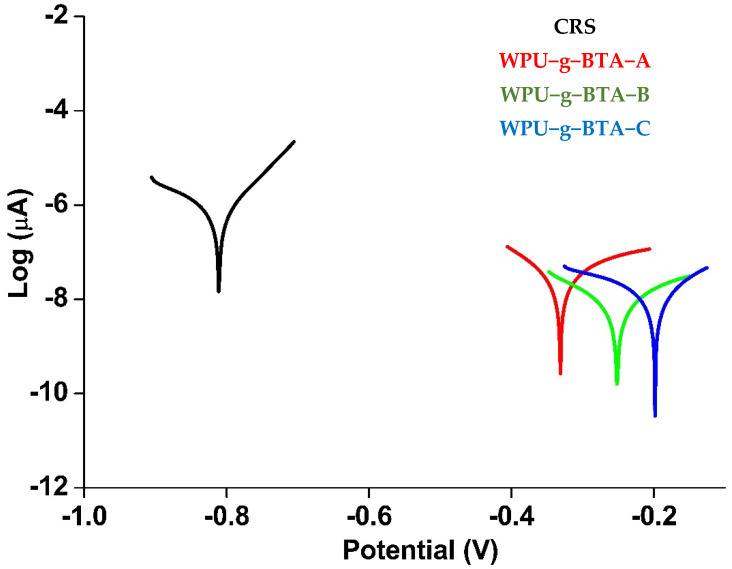
Tafel plots for raw CRS electrode and CRS electrode coated with WPU−g−BTA−A, WPU−g−BTA−B, and WPU−g−BTA−C, in 3.5 wt% NaCl_(aq)_ for 30 min immersion.

**Figure 4 molecules-27-07581-f004:**
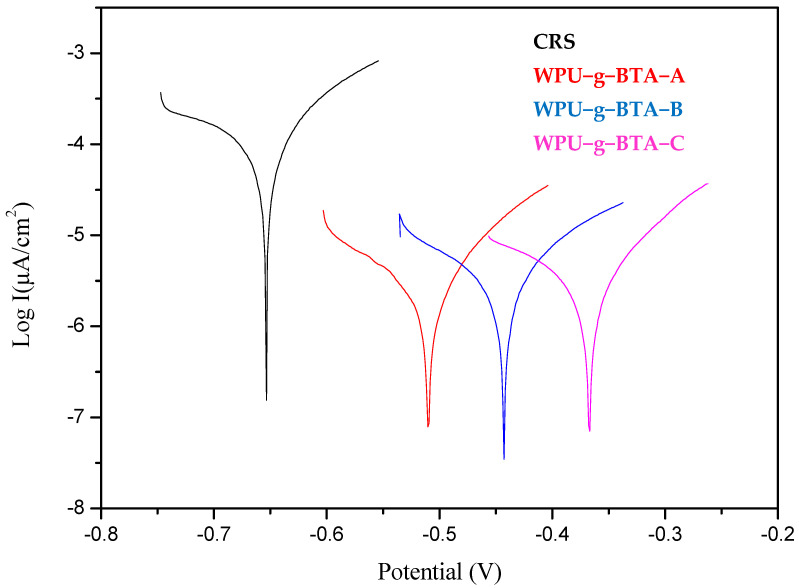
Tafel plots for raw CRS electrode and CRS electrode coated with WPU−g−BTA−A, WPU−g−BTA−B, and WPU−g−BTA−C in 3.5 wt% NaCl_(aq)_ for 3 days of immersion.

**Figure 5 molecules-27-07581-f005:**
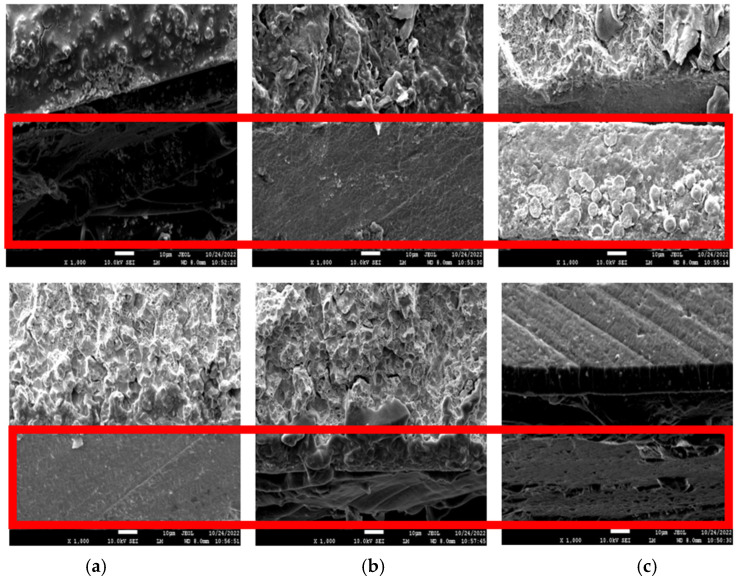
Cross-sectional SEM images of (**a**) WPU−g−BTA−A, (**b**) WPU−g−BTA−B, and (**c**) WPU−g−BTA−C in before corrosion (top) and after corrosion (bottom) in 3.5 wt% NaCl_(aq)_ after 3 days of immersion.

**Figure 6 molecules-27-07581-f006:**
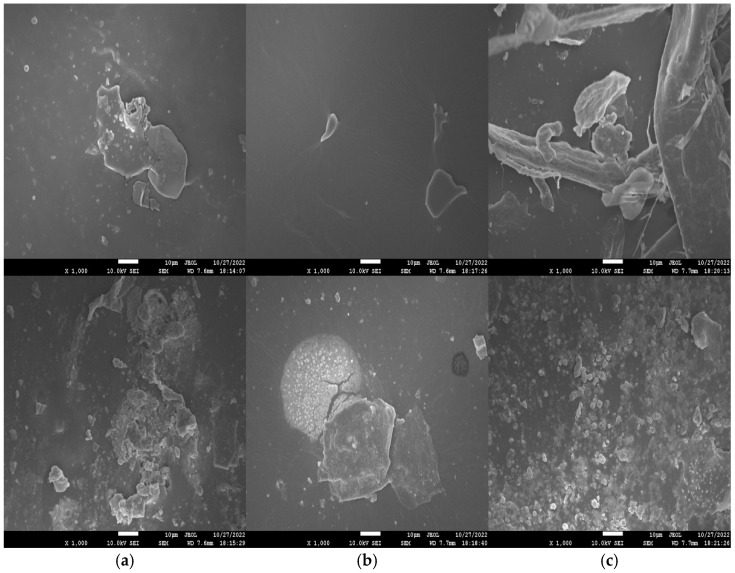
SEM images (top view) of (**a**) WPU−g−BTA−A, (**b**) WPU−g−BTA−B and (**c**) WPU−g−BTA−C before corrosion (top) and after corrosion (bottom) in 3.5 wt% s after 3 days of immersion.

**Figure 7 molecules-27-07581-f007:**
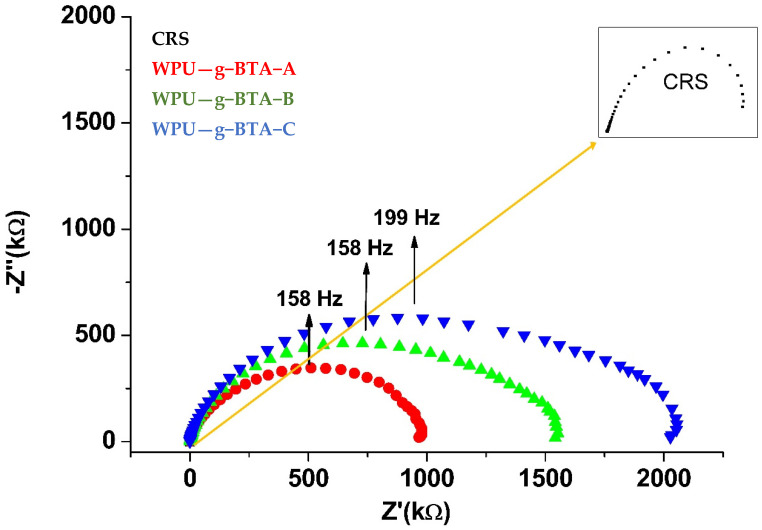
Nyquist plots for raw CRS electrode and CRS electrode coated with WPU−g−BTA−A, WPU−g−BTA−B, WPU−g−BTA−C in 3.5 wt% NaCl_(aq)_.

**Figure 8 molecules-27-07581-f008:**
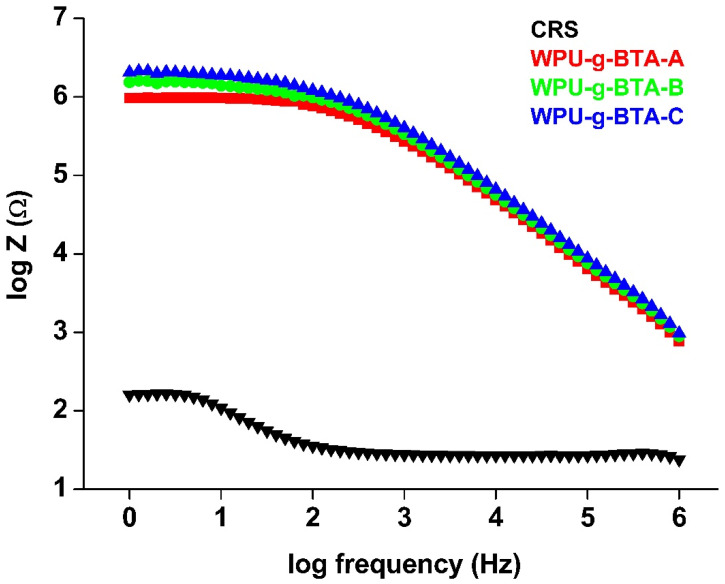
BODE plots for raw CRS electrode and CRS electrode coated with WPU−g−BTA−A, WPU−g−BTA−B, and WPU−g−BTA−C in 3.5 wt% NaCl_(aq)_.

**Figure 9 molecules-27-07581-f009:**
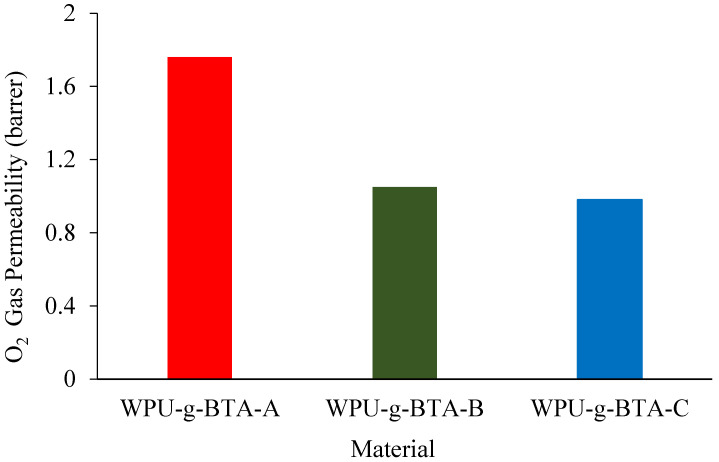
O_2_ permeability for CRS electrode coated with (red) WPU−g−BTA−A, (green) WPU−g−BTA−B, and (blue) WPU−g−BTA−C.

**Figure 10 molecules-27-07581-f010:**
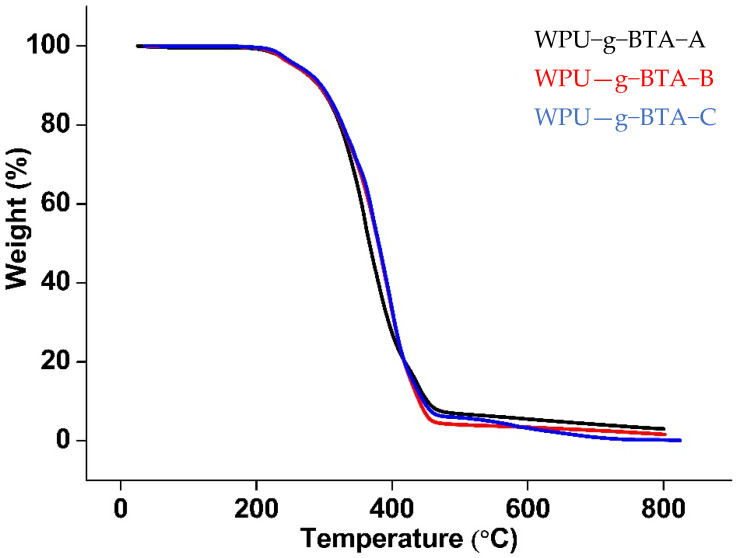
TGA curve of (black) WPU−g−BTA−A, (red) WPU−g−BTA−B, and (blue) WPU−g−BTA−C.

**Figure 11 molecules-27-07581-f011:**
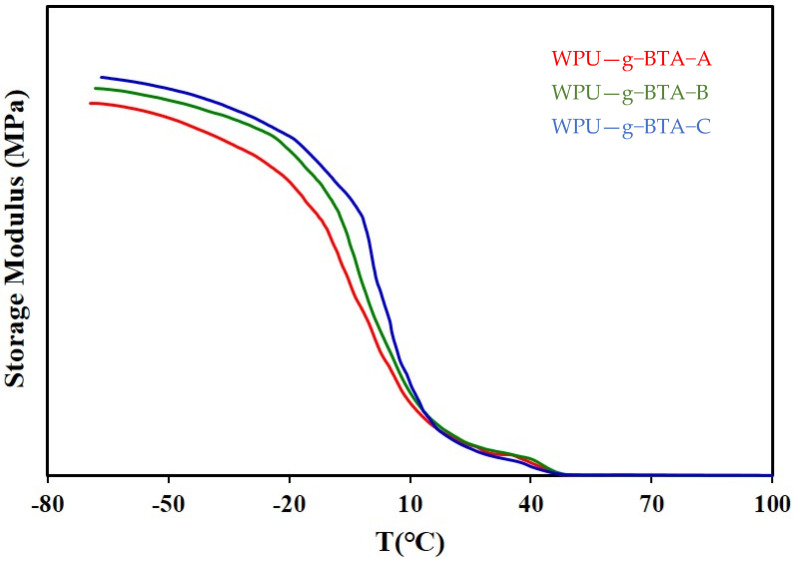
Storage modulus of (red) WPU−g−BTA−A, (green), WPU−g−BTA−A, and (blue) WPU−g−BTA−C.

**Figure 12 molecules-27-07581-f012:**
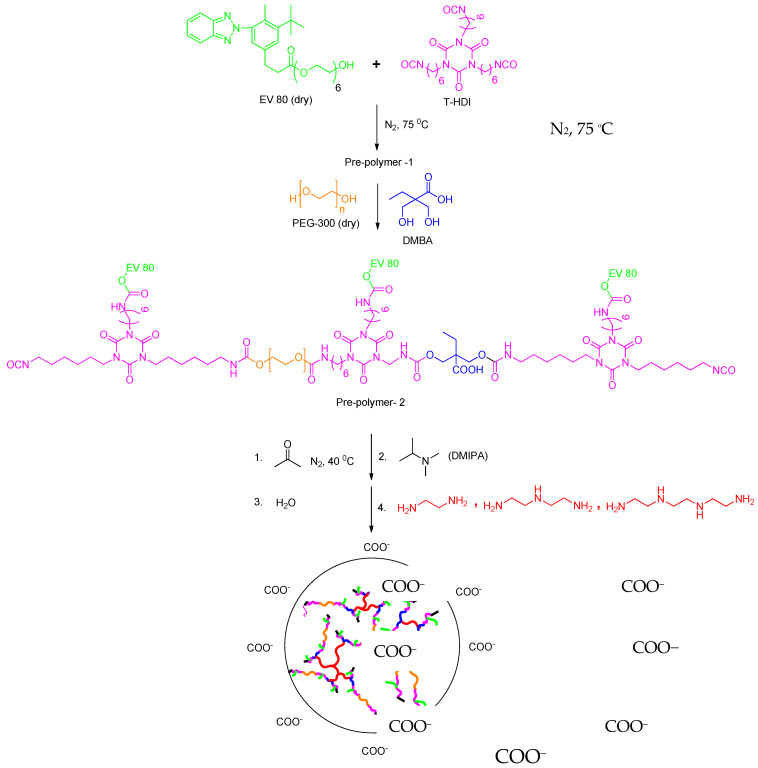
Synthetic route of WPU-g-BTA with side chain extenders.

**Table 1 molecules-27-07581-t001:** Gel content, O_2_ permeability, and thermal stability measured from TGA and DMA results of WPU-g-BTA coatings.

Sample Code		WPU-g-BTA-A	WPU-g-BTA-B	WPU-g-BTA-C
Gel content	Tube (g)	13.7608	13.7386	13.7784
	Sample (g)	0.6671	0.6675	0.6674
	THF (g)	20.171	20.0265	20.0255
	After drying (g)	13.8513	13.864	13.9231
	Gel Content (%)	13.57	18.79	21.68
Gas permeability	O_2_ (barrer)	1.758322	1.045648	0.984704
Thermal Stability	T_d(5%)_	252.75	257.47	257.47
	T_d(10%)_	288.26	292.41	295.69
	Char yield %	3	1.6	0.21
DMA	Stress (kgf/cm^2^)	51.19	47.56	50.41
	Elongation (%)	178.85	224.66	283.41
	100% modulus (kgf/cm^2^)	42.15	44.12	46.51

**Table 2 molecules-27-07581-t002:** Electrochemical parameters, the corresponding inhibition efficiency (IE %), EIS, and BODE results for raw CRS and CRS electrode coated with WPU-g-BTA-A, WPU-g-BTA-B, and WPU-g-BTA-C in 3.5 wt% NaCl _(aq)_ at 298 K.

Sample Code		CRS	WPU-g-BTA-A	WPU-g-BTA-B	WPU-g-BTA-C
**Coating thickness**	(μm)	60 ± 2	60 ± 2	60 ± 2	60 ± 2
**Electrochemical** **parameters**	Ecorr (V)	−0.81121	−0.3311	−0.25208	−0.19831
	icorr (μA/cm^2^)	71.773	0.079	0.068	0.020
	IE %	-	99.889	99.905	99.972
**EIS (Nyquist)**	Z’ (kΩ)	0.16	978.84	1552.69	2055.23
	-Z’’ (kΩ)	0.09	346.95	462.90	582.83
**BODE**	log(Z)	2.22	5.99	6.21	6.33

**Table 3 molecules-27-07581-t003:** Long term anti-corrosion study for raw CRS and CRS electrode coated with WPU-g-BTA-A, WPU-g-BTA-B, and WPU-g-BTA-C in 3.5 wt% NaCl (aq) at 298 K.

Sample Code	Ecorr (V)	icorr (µA.cm^−2^)	Rp (kΩ)	P_EF_ %
CRS	−0.634	4.98	2.587	-
WPU-g-BTA-A	−0.633	4.17	3.065	16.16
WPU-g-BTA-B	−0.576	4.08	2.349	18.13
WPU-g-BTA-C	−0.550	3.41	3.127	31.56

## Data Availability

Not applicable.

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
