# Peer review of "Preparation and Characterization of Water-borne Polyurethane Based on Benzotriazole as Pendant Group with Different N-Alkylated Chain Extenders and Its Application in Anticorrosion"

_molecules, 2022, doi:10.3390/molecules27217581_

Round 1
Reviewer 1 Report
Bibi et al. studied waterborne polyurethane based on benzotriazole as pendant group with different N-al-kylated chain extenders and its application in anticorrosion. The manuscript is innovative to a certain extent, but there are the following problems that need to be revised:
(1) The language in the manuscript needs to be greatly improved.
(2) The Introduction section is unattractive and authors are advised to refer to Journal of Colloid and Interface Science 506 (2017) 478-485 and Colloids and Surfaces A: Physicochemical and Engineering Aspects 645 (2022) 128892.
(3) Authors should carefully check the abbreviations in the manuscript to ensure the consistency of all abbreviations in the manuscript.
(4) It is suggested that the symbol of corrosion current density should be changed from Icorr to icorr.
(5) The X and Y axes of Nyquist plots should have equal values and some frequency values should be plotted.
(6) EIS data should be analyzed in more detail, and authors are advised to refer to Journal of Colloid and Interface Science and Journal of Colloid and Interface Science 609 (2022) 838–851.
(7) The synthetic part of the Synthesis of WPU-g-BTA should be described in more detail.
(8) The conclusion part suggests that the author presents it in points.
Author Response
Response to reviewer-1 1. The language in the manuscript needs to be greatly improved. The manuscript undergone an extensive English formatting service. 2. The introduction section is unattractive and authors are advised to refer to journal of colloids and interfaces sciences 506 (2017) 478-485 and colloids and surfaces A: physiochemical and engineering aspects 645 (2022) 128892. The introduction part was revised according the above mentioned references. 3. Authors should carefully check the abbreviations in the manuscript to ensure the consistency of all abbreviations in the manuscript. All abbreviations mentioned in the manuscript were revised. 4. It is suggested that the symbol of corrosion current density should be changed from Icorr to icorr. The symbol for corrosion current density changed from Icorr to icorr. 5. The X and Y axes of Nyquist plots should have equal values and some frequency values should be plotted. The figure 7 for Nyquist plot was revised. 6. EIS data should be analyzed in more detail, and authors are advised to refer to journal of colloid and interface science 609 (2022) 838-851. EIS data was revised according to the above mentioned reference. 7. The synthetic part of the synthesis of WPU-g-BTA should be described in more detail. The synthetic part was revised. 8. The conclusion part suggests that the author present it in points. The conclusions presented in points.

Reviewer 2 Report
-Figure 9 is not clear
-Surface microscopy is not provided
-Long term corrosion experiments to investigate the integrity of the coating is not provided
-Report SEM of the coating as is.
-Report SEM micrographs after immersion of at least 24 hours in an aggressive media such as 3.5% NaCl.
-Do a cross section SEM to check the coating adhesion.
Author Response
Response to reviewer-2
- Figure 9 is not clear.
A high resolution Figure 12 (previously figure 9) was provided.
- Surface microscopy is not provided.
Surface microscopy (cross sectional and top view) is provided as figure 5 and figure 6.
- Long term corrosion experiments to investigate the integrity of the coating is not provided.
Long term corrosion experiment was studied for three days and results are shown in figure 4 and table 3.
- Report SEM of the coating as is.
SEM is reported in Figures 5 & 6.
- Report SEM micrographs after immersion of at least 24 hours in aggressive media such as 3.5 wt% NaCl.
SEM micrographs of the as-prepared coatings are reported in Figures 5 & 6 before and after corrosion respectively.
- Do a cross-section SEM to check the coating adhesion.
Cross-sectional images of as-prepared coatings are given in Figures 5 & 6 before and after corrosion.

Round 2
Reviewer 1 Report
I have read the revised version of the Manuscript and found that the authors have taken into accounts the concerns that I raised. Thus I recommend it for publication.
Reviewer 2 Report
Great work